# Research

behaviour, ecology, environmental science

beaked whales, mid-frequency active sonar, stranding event, Navy sonar, *Ziphiidae*

**Author for correspondence:**
Anne E. Simonis
e-mail: anne.simonis@noaa.gov

# Co-occurrence of beaked whale strandings and naval sonar in the Mariana Islands, Western Pacific

Anne E. Simonis[1], Robert L. Brownell Jr[3], Bruce J. Thayre[4], Jennifer S. Trickey[4], Erin M. Oleson[2], Roderick Huntington[4,5] and Simone Baumann-Pickering[4]

[1]Contractor to Pacific Islands Fisheries Science Center, and [2]Pacific Islands Fisheries Science Center, National Oceanic and Atmospheric Administration, Honolulu, HI, USA
[3]Southwest Fisheries Science Center, National Oceanic and Atmospheric Administration, Monterey, CA, USA
[4]Scripps Institution of Oceanography, UCSD, La Jolla, CA, USA
[5]Mount Edgecumbe High School, Sitka, AK, USA

AES, 0000-0002-8026-6233

Mid-frequency active sonar (MFAS), used for antisubmarine warfare (ASW), has been associated with multiple beaked whale (BW) mass stranding events. Multinational naval ASW exercises have used MFAS offshore of the Mariana Archipelago semi-annually since 2006. We report BW and MFAS acoustic activity near the islands of Saipan and Tinian from March 2010 to November 2014. Signals from Cuvier's (*Ziphius cavirostris*) and Blainville's beaked whales (*Mesoplodon densirostris*), and a third unidentified BW species, were detected throughout the recording period. Both recorders documented MFAS on 21 August 2011 before two Cuvier's beaked whales stranded on 22–23 August 2011. We compared the history of known naval operations and BW strandings from the Mariana Archipelago to consider potential threats to BW populations. Eight BW stranding events between June 2006 and January 2019 each included one to three animals. Half of these strandings occurred during or within 6 days after naval activities, and this co-occurrence is highly significant. We highlight strandings of individual BWs can be associated with ASW, and emphasize the value of ongoing passive acoustic monitoring, especially for beaked whales that are difficult to visually detect at sea. We strongly recommend more visual monitoring efforts, at sea and along coastlines, for stranded cetaceans before, during and after naval exercises.

## 1. Introduction

Beaked whales (Family Ziphiidae) are a poorly understood family of 23 species of deep-diving cetaceans. Beaked whales compared to other cetacean species, are reported to be more vulnerable to severe and sometimes fatal responses to mid-frequency active sonar (MFAS) operations [1–5]. Since the introduction of MFAS in the range of 4.5–5.5 kHz in the early 1960s, there have been at least 12 beaked whale mass stranding events (involving two or more individuals) that coincided in space and time with naval exercises that may have used MFAS [6]. An additional 27 other beaked whale mass stranding events have been documented near a naval base or ship, but very few have had direct evidence of associated sonar use [6]. Filadelfo *et al.* [7] used the same stranding data as D'Amico *et al.* [6], with more robust information on historical naval activity to examine the correlation between beaked whale mass strandings and military events. The author's conclusions were that beaked whale mass strandings were correlated with naval activity in the Mediterranean and Caribbean Seas, but not correlated off the coasts

of Japan and southern California. However, these authors only included data beginning in 1978 for Japan, and did not consider atypical mass beaked whale stranding events that occurred during the 1960s and 1970s in Japan, which may have also been associated with MFAS [8].

The Mariana Archipelago, consisting of the islands of Guam to the south and the Commonwealth of the Northern Mariana Islands (including Saipan and Tinian, hereafter referred to as the Northern Mariana Islands) to the north, has been designated as a strategic location by the US Department of Defense, and serves as the principal US military training and basing location in the Western Pacific. Until recently, the distribution and abundance of cetaceans in the Mariana Archipelago was relatively unstudied. Since 1993, marine mammal strandings in the Northern Mariana Islands have been documented and archived by the Department of Lands and Natural Resources Division of Fish and Wildlife (DFW), mainly from Saipan. Additional stranding records from Guam, have been collected since 1962 by the Department of Agriculture, Division of Aquatic and Wildlife Resources (DAWR). Historical marine mammal strandings included a variety of cetacean species [9–14], with the first beaked whale stranding recorded on Guam in 2007 (single Cuvier's beaked whale; *Ziphius cavirostris*), followed by two additional stranding events of single Cuvier's beaked whales in 2008. Between 2015 and 2019, there were four documented strandings of Cuvier's beaked whales involving one group of two or three animals (one live-stranded animal may have re-stranded dead later), and three other single animals (B. Tibbatts and K. West 2011–2019, personal communications with R.L.B.). In the Northern Mariana Islands, the only records of beaked whale strandings include two Cuvier's beaked whales that stranded in August 2011 on the west coast of Saipan. Details on beaked whale strandings from the Mariana Archipelago from August 2007 to January 2019 are included in table 1.

Visual and acoustic monitoring efforts of marine mammals in the region are ongoing to improve the understanding of the distribution, abundance and effectiveness of mitigation measures for marine mammals impacted by military activities in the Mariana Island Range Complex. Visual surveys since 2007 have documented Cuvier's, Blainville's (*Mesoplodon densirostris*) and unconfirmed *Mesoplodon* spp. beaked whales in deep waters (greater than 650 m) [21]. Since 2010, acoustic monitoring has documented echolocation clicks from Cuvier's, Blainville's and an unidentified beaked whale (possibly the ginkgo-toothed whale, *M. ginkgodens* characterized as 'BWC' by [22]) near Saipan and Tinian throughout the year [23–27].

The purpose of this study was to document the seasonal acoustic presence of beaked whales near Saipan and Tinian using high-frequency acoustic recording packages (HARP) [28]. After two Cuvier's beaked whales stranded on Saipan in 2011 during our study/recording period, we searched the acoustic data for MFAS used in antisubmarine warfare (ASW) operations. We document the acoustic activity of beaked whales and MFAS over 2010–2014, and also reviewed unclassified, publicly available reports of multinational ASW activities over the longer time period of 2006–2019. We then prepared a timeline with these naval activities and the HARP recordings, and compared them to the reported beaked whale strandings from the Mariana Archipelago.

## 2. Results

### (a) High levels of beaked whale acoustic activity

Throughout 2010–2014, three different beaked whale signal types were detected, produced by Blainville's and Cuvier's beaked whales, and the 'BWC' signal. Beaked whale signals were detected during 94% and 80% of all weeks with recording effort at the West (15° 19.026′ N, 145° 27.463′ E) and East (15° 2.344′ N, 145° 45.130′ E) HARPs, respectively (figures 1 and 2). Blainville's beaked whale signals were the most frequent beaked whale signal type observed at both sites, detected on 35% of recording days for an average 4.8 min per day on the West HARP and 28% of days for an average 5.8 min on the East HARP. Cuvier's beaked whale signals were detected on 19% of recording days for an average 1.2 min per day on the West HARP, compared to only 7% of recording days for an average 0.1 min per day at the East HARP. There were no Cuvier's beaked whale signals detected at the East HARP during January to November 2014 (figure 2). The 'BWC' signal type was similarly present at both sites during 11% and 5% of recording days, with average daily durations of 2.1 and 0.3 min at the West and East HARPs, respectively.

### (b) Presence of military sonar (2010–2014)

During 2010, there were no detections of MFAS throughout the recordings from the West HARP and no recordings were available from the East HARP. MFAS events were detected on a total of 35 days between 2011 and 2014, with MFAS events lasting from 1 to 18 days (table 2). When MFAS packets were detected, they generally occurred in consecutive bouts with fewer than 2 min between sonar packets, followed by 'breaks' in sonar activity, which ranged from 15 min to nearly 3 h (table 2).

### (c) Association of beaked whale stranding events with ASW training (2006–2019)

We found public reports of 21 scheduled or completed multinational naval ASW exercises around the Mariana Archipelago (figure 3; electronic supplementary material, table S1). The timeline in figure 3 shows a list of known beaked whale strandings and reported US Naval joint exercises around the Mariana Archipelago that included ASW. Multinational ASW exercises were reported as early as 2003, but began to occur semi-annually in 2006. The US Navy reported four major international antisubmarine operations during the active HARP recording periods. One of these events (Valiant Shield V: 15–23 September 2014) was detected acoustically on 15–16 September and 21–22 September, with MFAS also detected during 7 days prior and 5 days following the respective start and end dates of the operation. MFAS was also detected on 17 days that were not included in publicly reported events.

Between August 2007 and January 2019, there were eight stranding events of one to three Cuvier's beaked whales, totalling 10 or 11 individuals (one live animal that was returned to sea may have been the same animal that re-stranded dead later). In relation to a stranding event in August 2011, the US Navy Mariana Islands Testing and Training Environmental Impacts Statement (EIS) reported that there were no US Navy testing or training activities in the days prior to the

**Table 1.** Beaked whale strandings from August 2007 to January 2019 within the Mariana Archipelago.

| date | species | Island | stranding location | number of individuals | total length | sex | outcome | carcass examined | notes | references |
|---|---|---|---|---|---|---|---|---|---|---|
| 30 Aug 2007 | Zc | Guam | Piti Bay, Cabras Island | 1 | n.a. | male | unknown; pushed back to sea | no | sex based on photo of erupted teeth | K. West 12 Feb 2019, personal communication; B. Tibbatts 6 Sep 2007, 11 Sep 2007, personal communication |
| 27 Jan 2008 | Zc | Guam | Luminao Reef | 1 | 520 cm | male | found dead | yes—tissues samples collected | specimen was badly decomposed; skull collected | Brown [15]; B. Tibbatts 29 Jan 2008, personal communication |
| 19 Jul 2008 | Zc | Guam | Dadi/Rizal Beach | 1 | n.a. | n.a. | unknown; last seen in shallow water | no | near-stranding: animal was seen in shallow water (<100 ft depth) near shore during aerial survey | B. Tibbatts 1 Aug 2008, personal communication |
| 22 Aug 2011 | Zc | Saipan | Oleai Beach | 1 | n.a. | female | found dead | yes—head frozen, samples taken from brain, eye and lymph nodes | first day of stranding event; first reported at 10.30 | West et al. [16]; Kuam News [17] |
| 23 Aug 2011 | Zc | Saipan | Micro Beach | 1 | 439 cm | male | euthanized | yes—gross necropsy | live stranding; second day of stranding event; first reported at 08.00 | West et al. [16] |
| 23 Mar 2015 | Zc | Guam | Bile Bay | 2–3 | 480 cm | male | 1 found dead; 1–2 unknown | yes—tissue samples and stomach contents | two whales stranded live in different locations were refloated; after, one whale was found dead which may have been a third whale or one of the first two | Kuam News [18]; B. Tibbatts 4 Mar 2019, personal communication |
| 26 Jul 2015 | Zc | Guam | Agat Reef | 1 | 516 cm | male | found dead | yes—tissue samples and stomach contents | specimen was badly decomposed | West et al. [16]; B. Tibbatts 27 Jul 2015, personal communication |
| 8 Mar 2016 | Zc | Guam | Agat Bay | 1 | ca 366[1] cm | n.a. | unknown; herded out to open water | n.a. | reported in shallow water at 08.10 | Pacific Daily News [19] |
| 17 Jan 2019 | Zc | Guam | Agat | 1 | 338 cm | male | euthanized | yes—tissue samples and stomach contents | live stranding; restranded on Dadi Beach where it was euthanized | K. West 13 Feb 2019, personal communication; Kuam News [20] |

Proc. R. Soc. B 287: 20200070

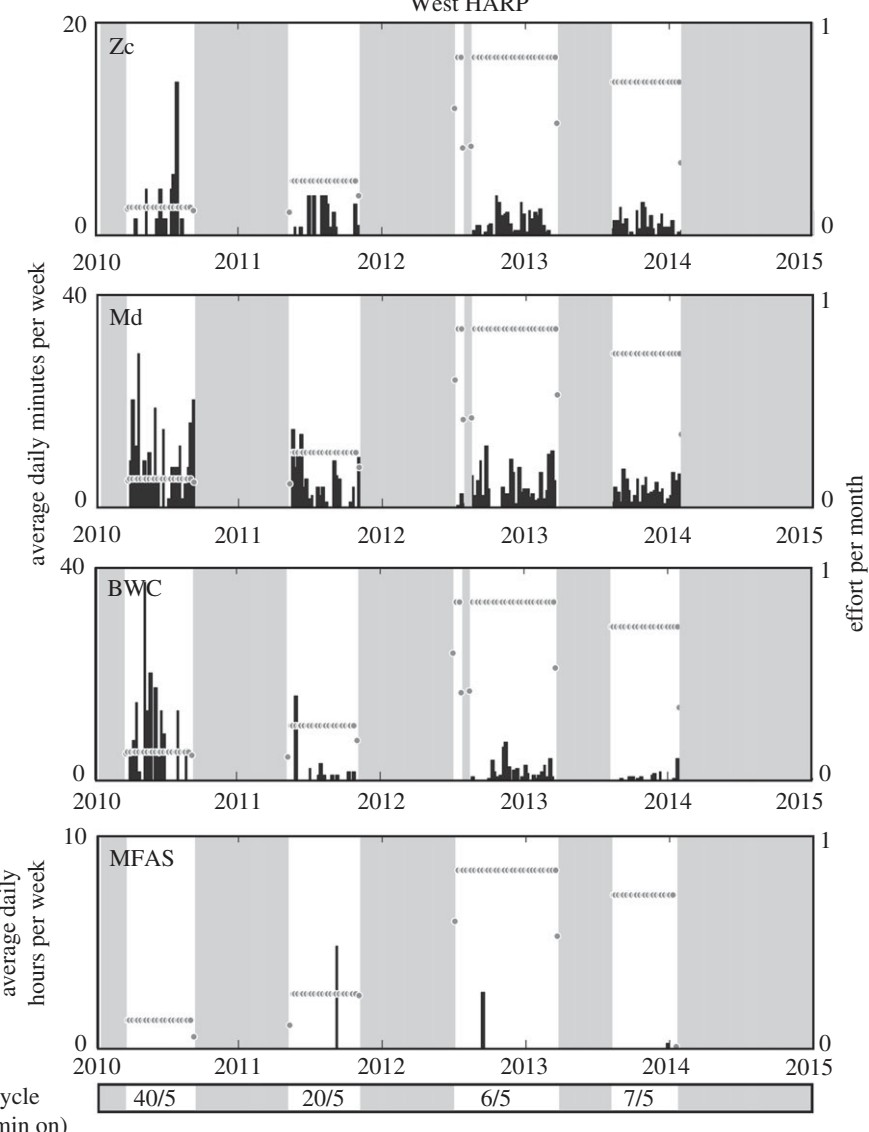

**Figure 1.** Acoustic presence of Cuvier's (Zc), Blainville's (Md) and 'BWC' beaked whale signal types and mid-frequency active (MFA) sonar at the West HARP from 2010 to 2014. Black bars indicate weekly averages of the daily minutes for beaked whale presence and daily hours for MFAS presence, both effort-adjusted. Effort per week is plotted with grey circles to indicate the recording schedules associated with each time period. The presence of all analyst detections of MFAS events is shown here, regardless of received level or data quality. Grey boxes indicate times of no recording effort. Duty cycle (minutes in period/recording minutes) is shown in the bottom panel.

stranding event [30]. However, the US Navy has recently confirmed that there was sonar use during unit-level training (in an unnamed exercise) within 72 h and within 80 nmi of the stranding event on 22–23 August 2011 (US Pacific Fleet N465, 4 March 2019, personal communication). Assuming a conservative number of total individuals, six of the 10 Cuvier's beaked whales, from four of eight events, stranded during or within 6 days of a naval ASW exercise.

We used a simulation to investigate the probability that four of the eight beaked whale stranding events occurred with navy events by chance. Eight random days were drawn from the entire observation period to simulate separate stranding events. We considered simulated stranding events as associated with naval events if they occurred during, or within 6 days after a naval event. In 10 000 random draws, the median number of simulated stranding events that were associated with naval events was 1 (mean ± s.d. = 0.49 ± 0.68), and the probability that four of eight stranding events were randomly associated with naval events was 0.1% (electronic supplementary material, table S2). The naval event/stranding

event association window represents only 6.1% of the total observation period (293 days of 4771 days observed). This underscores the small probability of any stranding event occurring within the association window, especially four of eight observed stranding events.

## 3. Discussion

### (a) Beaked whales

The acoustic record indicates that the habitats near both recording locations are used by Blainville's, Cuvier's and an unidentified beaked whale that produces the 'BWC' signal type [22]. The West and East HARP locations may be considered as potentially important beaked whale habitat, given that beaked whales were present in 94% and 80% of the weeks with recording effort at each respective location. Although not modelled in this study, the detection range of beaked whale signals at each HARP is likely to be limited [31,32] and estimated to be less than 5 km, given the high-frequency content of beaked

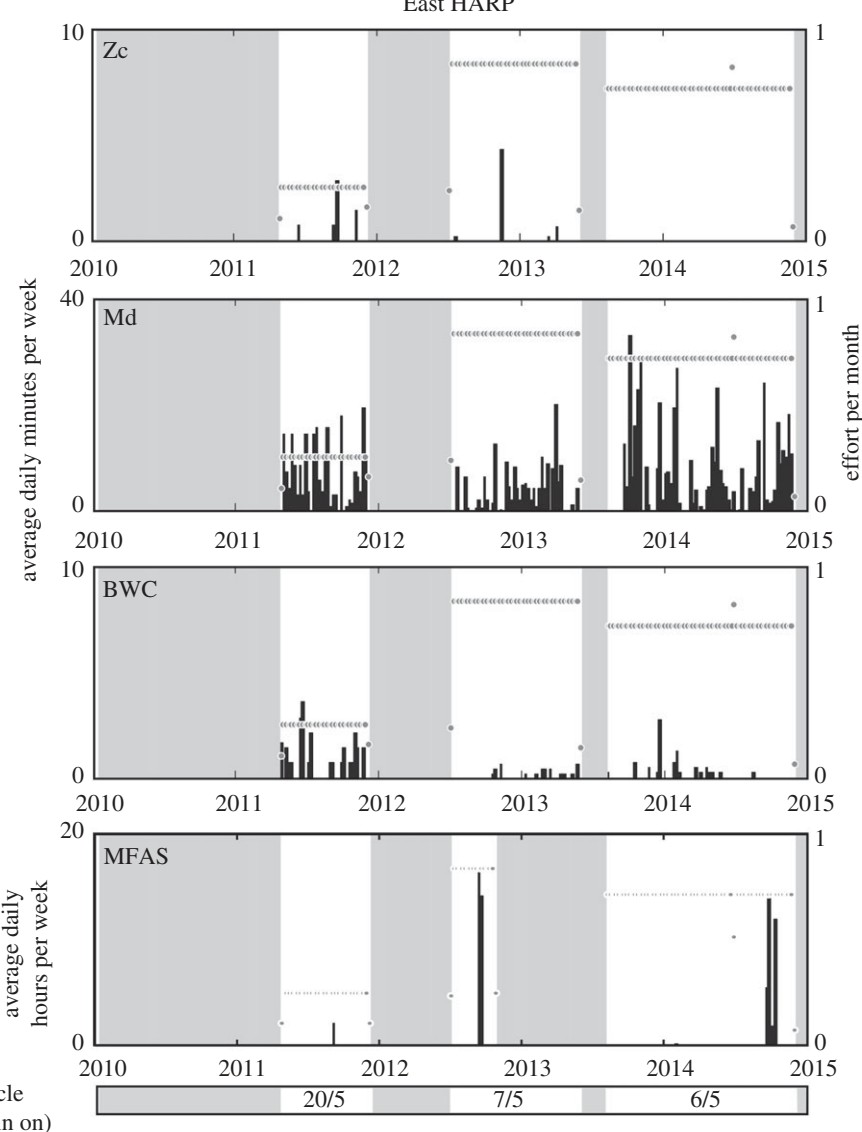

**Figure 2.** Acoustic presence of Cuvier's (Zc), Blainville's (Md) and 'BWC' beaked whale signal types and mid-frequency active (MFA) sonar at the East HARP from 2010 to 2014. Black bars indicate weekly averages of the daily minutes for beaked whale presence and daily hours for MFAS presence, both effort-adjusted. Effort per week is plotted with grey circles to indicate the recording schedules associated with each time period. The presence of all analyst detections of MFAS is shown here, regardless of received level or data quality. Grey boxes indicate times of no recording effort. Duty cycle (minutes in period/recording minutes) is shown in the bottom panel. Reduced effort for MFAS signals in 2013 was the result of a failure of the instrument that affected frequencies less than 5 kHz.

whale echolocation clicks. Another indicator of the low probability of detecting beaked whales in the area is the consistently low number of minutes per day with detections. As such, the absence of beaked whale signals in a recording cannot be broadly interpreted as absence in the greater area, but their presence can provide an indication of relative occurrence rates and seasonal fluctuations in occurrence. The different detection rates of each BW signal is likely related to different habitat conditions at each location. In particular, the low occurrence rates for the 'BWC' signal are likely to be related to low detection ranges associated with the low source level of the signal, based on the very broad bandwidth, low received levels and short encounters observed at all recording locations across the North Pacific [22].

## (b) Presence of military sonar (2010–2014)

MFAS was detected each year from 2011 to 2014, including one day preceding a beaked whale stranding event on Saipan, a

location where beaked whale strandings had not been previously recorded. The range of MFAS received levels at the recording locations (89–132 $dB_{RMS}$ re: 1 μPa), including the day preceding the 2011 stranding event, were within the range of received levels shown to elicit moderate to strong avoidance responses in beaked whales during controlled exposure experiments (89–140 $dB_{RMS}$ re: 1 μPa [2,33–36]). The highest received levels were recorded on the West HARP, which was nearest to the location of the 2011 stranding (west coast of Saipan). However, beaked whales in the broader area, including the beaked whales that stranded, may have been farther from or closer to the source and experienced lower or higher levels of MFAS, respectively. Multiple sonar packets at different frequencies and at varying received levels often occurred simultaneously (see diagram in electronic supplementary material, figure S1). Although the sonar type and position of ships that emitted these signals are unknown, these observations suggest the presence of multiple sources (ships) at different locations. We conclude that the presence

**Table 2.** Acoustic characterization of MFAS packets detected at West and East HARP locations from 2011 to 2014 (no MFAS was detected in 2010). MFAS packets include groups of frequency-upsweep, downsweep and tonal pulses that occur within 5 s of each other. Encounter durations are shown in hours and minutes (HH.MM), with mean (top) and range (bottom) shown for years with multiple encounters. The inter-packet interval is the time between the end of one sonar packet and the beginning of the next, omitting breaks (gaps greater than 10 min). Received level (RL) is reported as peak-to-peak (PP) and root-mean-square (RMS) values. Sound exposure level (SEL) is the cumulative sum-of-square pressures over the duration of a sonar packet. For each metric, mean ± s.d. are shown in bold and 10th, 50th, 90th percentiles are shown in parentheses. Note that the number of described packets does not equal total packets due to duty cycled data and peak-to-peak received level threshold (115 dB re: 1 μPa).

| | 2011 | | 2012 | | 2013 | 2014 |
| --- | --- | --- | --- | --- | --- | --- |
| | West | East | West | East | West | East |
| dates with MFAS | 21 Aug | 21 Aug | 28 Aug–1 Sep 21 Sep | 27 Aug–9 Sep 21 Sep | 16 Dec | 16 Jan 8–17, 21–22, 24–28 Sep |
| number of packets | 215 | 89 | 257 | 949 | 51 | 668 |
| encounter duration (HH.MM) | 08.18 | 03.41 | 01.06 (00.06–04.47) | 2.05 (00.01–14.42) | 01.05 | 8.50 (00.43–18.50) |
| packet duration (s) | 2.0 ± 0.5 (1.4 2.0 2.6) | 2.1 ± 0.5 (1.2 2.3 2.7) | 2.4 ± 0.4 (1.8 2.5 2.7) | 2.0 ± 0.4 (1.5 2.0 2.6) | 2.4 ± 0.5 (1.8 2.7 2.8) | 6.0 ± 2.8 (1.7 7.2 8.9) |
| packet interval (s) | 40.2 ± 140.4 (6.7 15.0 41.4) | 58.4 ± 181.4 (7.5 22.1 51.5) | 106.9 ± 63.7 (12.4 97.7 193.8) | 92.2 ± 57.2 (30.1 96.5 192.1) | 49.3 ± 33.7 (39.2 40.1 57.4) | 69.4 ± 59.4 (12.2 49.9 192.8) |
| RL PP (dB re: 1 μPa) | 136.3 ± 9.9 (123.7 135.4 149.6) | 123.7 ± 4.8 (117.4 123.5 129.4) | 119.9 ± 4.3 (115.4 118.6 125.9) | 120.3 ± 4.7 (115.5 118.8 127.6) | 120.8 ± 4.8 (115.6 120.4 127.0) | 123.0 ± 6.4 (116.4 121.2 133.0) |
| RL RMS (dB re: 1 μPa) | 118.0 ± 10.6 (104.4 116.5 132.2) | 104.3 ± 5.9 (97.2 103.4 112.1) | 98.8 ± 3.5 (95.8 98.0 102.5) | 102.2 ± 5.5 (96.4 101.1 110.3) | 100.7 ± 4.7 (96.4 99.2 105.7) | 101.9 ± 6.3 (96.4 99.9 111.6) |
| SEL (dB re: 1 μPa²-s) | 120.9 ± 9.9 (108.1 119.3 134.7) | 107.4 ± 5.1 (101.5 106.3 113.4) | 102.5 ± 2.9 (99.8 101.9 105.8) | 105.1 ± 5.0 (99.9 103.7 112.8) | 104.4 ± 3.8 (100.7 103.5 108.4) | 108.9 ± 5.1 (103.4 107.8 116.1) |

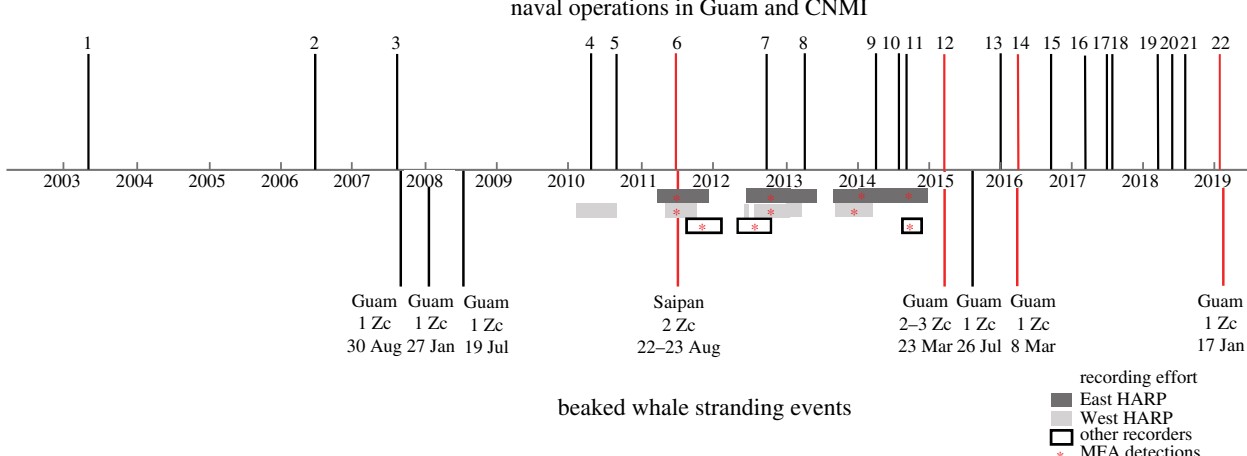

**Figure 3.** Timeline of beaked whale strandings on Guam and Northern Mariana Islands with publicly reported major multinational naval training operations in the Mariana Islands Range Complex from 2003 to 2019. Sonar-associated beaked whale stranding events and naval operations are shown with red lines. Acoustic recording effort is shown for the East HARP (light grey), West HARP (dark grey), and other recording effort published by Munger *et al.* [25] and Klinck *et al.* [26,27] is shown in boxes with no fill. Deployments with detections of MFAS are marked with asterisks. Details of the numbered naval operations are described in the electronic supplementary material, table S1. (Online version in colour.)

of multiple MFAS sources in beaked whale habitat may have contributed to this stranding event.

The duty cycled recordings limited the detection of events with durations less than 35 min in 2010, 15 min in 2011 and 1–2 min in 2012–2014. The majority of MFAS encounters that we observed had durations longer than 1 h, suggesting the duty cycle did not significantly limit MFAS detection overall. The duty-cycled nature of the recordings results in an incomplete report of acoustic activity at the recording locations; however, the signal characteristics reported here should be representative of the signals that were not recorded.

### (c) Association of beaked whale stranding events with MFAS (2006–2019)

Since 2007 there has been a strong association between beaked whale stranding events with the presence of multinational naval ASW training operations. No beaked whale strandings were reported from the Mariana Archipelago between 1962 and 2006, but from 2007 to January 2019, eight beaked whale stranding events (10–11 individuals) have been reported on Guam and Saipan, with 50% (four of eight) of the events associated with reported naval operations (figure 3). The 2011 beaked whale stranding on Saipan was not associated with a publicly reported exercise, but MFAS was detected on the HARPs prior to the stranding and the US Navy confirmed the use of sonar during a unit-level training exercise on 21 August 2011 in an area 80 nmi from the beaked whale stranding location on 23 August (US Pacific Fleet N465, 4 March 2019, personal communication). The US Navy has also confirmed that the MFAS used in major multinational naval ASW training exercises was associated with the beaked whale stranding events in March 2015 and March 2016 (US Pacific Fleet N465, 4 March 2019, personal communication). The US Navy only has the responsibility to report major (multinational) sonar and training exercises [37], and sonar used outside of these training operations is not usually public knowledge.

Previous studies suggest that 9% of global beaked whale mass strandings are associated with naval operations

involving MFAS [6], but by only considering mass strandings (two or more animals, excepting mothers with dependent calves), this is a conservative metric; single animal strandings may also be associated with MFAS. In the Mariana Archipelago, six stranding events between 2007 and early 2019 included a single animal, and two of those six were associated with naval operations, suggesting single animal strandings merit careful examination. Expanding the consideration of sonar-associated strandings to include events with single animals, we report 50% of beaked whale stranding events in the Mariana Archipelago associated with ASW and MFAS activity. The high association (50%) of beaked whale stranding events with ASW and sonar activity, with the relative lack of beaked whale strandings before 2007, suggest that there may be high risks of sonar-associated beaked whale strandings in the Mariana Archipelago.

### (d) Risks for sonar-associated strandings

Infrequent and unpredictable noise is often perceived as a threat [38] and compared to a naive animal, stronger or weaker reactions to noise may result from habituation or associative learning [39]. During 2011–2014 in the Northern Mariana Islands, acoustic detections of sonar events were infrequently recorded, including 1 day in 2011, 15 days in 2012, 1 day in 2013 and 18 days in 2014. Other authors indicate similarly infrequent MFAS in the Mariana Archipelago, during and outside of documented training exercises [26] (figure 3). Especially in a pristine acoustic environment, beaked whales have shown strong avoidance responses to both near and distant MFAS [36]. Conversely, after decades of exposure to MFAS disturbances, some resident beaked whales near navy ranges may habituate to sonar or learn to abandon preferred habitat during MFAS operations [3,33,40]; however, there may still be high energetic costs associated with avoiding MFAS [41]. In the waters surrounding the Mariana Archipelago, the infrequent sonar activity, in conjunction with quiet ambient noise levels [42], may increase the severity in the behavioural response of beaked whales to sonar compared to populations living with higher ambient noise levels or those which have become

habituated to frequent MFAS activity. The risk for sonar-associated strandings may be similarly high in other regions with similar conditions.

When one of the two 2011 Saipan beaked whales was examined (a 4.39 m male), a heavy infestation of giant nematodes (*Crassicauda* sp.) was observed in both kidneys (K. West 13 February 2019, personal communication). The US Navy 2019 EIS for the Mariana Islands Range Complex suggested that this heavy parasite load could be a potential factor leading to the stranding, because the whale was already compromised [43]. However, these nematodes are observed in most dead beaked whales, regardless of the cause of death. They are usually found in healthy beaked whales taken by Japanese whalers (R.L.B. 2019, unpublished data) and stranded beaked whales (single and mass strandings) of various species, including Cuvier's beaked whales [1,29,44]. Therefore, we believe that MFAS, and not these commonly occurring parasites, was the primary factor in relation to this stranding event.

Looking into the future, optimal investigations of beaked whale behaviour and MFAS using passive acoustic monitoring should incorporate a high density of acoustic sensors in a variety of habitats, capable of recording continuously over multiple seasonal cycles. Consistent stranding networks are needed to monitor and respond to individual and mass strandings in time to investigate the hypotheses associated with the causes of stranding events, including acoustic-barotrauma [1]. Ideally, full disclosure of the timing and position of MFAS events would support more robust assessments of the potential risk for sonar-associated strandings.

## 4. Conclusion

The acoustic activity of three beaked whale species was regularly detected in the Northern Mariana Islands between 2010 and 2014, indicating this is an important habitat for beaked whales. While MFAS was infrequently detected, here we report a sonar event in 2011 that was associated with the stranding of two Cuvier's beaked whales on Saipan, along with three other beaked whale stranding events in 2015, 2016 and 2019, that were associated with major multinational ASW exercises, adding the Mariana Archipelago to a global list of locations, including the Bahamas, Canary Islands and Mediterranean (Italy and Greece), where sonar-associated beaked whale strandings have been documented. The sonar-associated with the 2011 Saipan stranding event was not linked with a publicly reported major (multinational) ASW operation, suggesting that other sonar-associated strandings may be underestimated. In addition, we have shown for the first time several single beaked whale strandings that were associated with major (multinational) ASW training events, indicating that strandings of individual animals should be considered as potentially sonar-associated. Passive acoustic monitoring continues to be a valuable tool to document the presence of visually cryptic beaked whales as well as naval sonar activity. Acoustic monitoring should be combined with the recommendation of Filadelfo *et al.* [7] that 'the fullest documentation of all stranding events is warranted' before, during and after future naval exercises throughout the Mariana Archipelago. Additional effort is also needed to improve the capacity to respond to and investigate (necropsy) any sonar-associated strandings to determine their cause.

## 5. Material and methods

### (a) Acoustic data collection

Acoustic recordings were collected at a sampling rate of 200 kHz at two locations near the islands of Saipan and Tinian from 2010 to 2014 from High-Frequency Acoustic Recording Packages (HARPs; [28]). All instruments were bottom-mounted and deployed to seafloor depths of 600–700 m for the 'West HARP' location (15° 19.026′ N, 145° 27.463′ E), and 1000 m at the 'East HARP' location (15° 2.344′ N, 145° 45.130′ E). From 2011 to 2013, the temporal coverage of recordings overlapped at both locations; however, recordings were not collected in all months for all years (figures 1 and 2). The hydrophone used was an omni-directional sensor (ITC-1042, International Transducer Corporation, Santa Barbara, CA), which had an approximately flat (±2 dB) hydrophone sensitivity from 10 Hz to 100 kHz of −200 dB re V/µPa. Each system contained a custom-built preamplifier board and bandpass filter [28]. The calibrated system response was accounted for during the analysis. The ability to assess the presence of beaked whales varies as a function of the recording schedule, the location and the relative abundance and vocal activity of the beaked whale species of interest [45]. Accordingly, the duty-cycled recording schedules used in this study may result in an underestimation of actual beaked whale presence, and the different recording schedules used throughout the study should be considered when evaluating relative abundance of acoustic activity.

### (b) Beaked whale detection and classification

The acoustic activity of beaked whale signals was detected using a multistep detection process following methods described in Baumann-Pickering *et al.* [46]. All echolocation clicks were detected using a computer algorithm [47]. Click detections were then classified as Cuvier's, Blainville's or 'BWC' signal types based on the spectral and temporal characteristics of the species-specific descriptions provided by Baumann-Pickering *et al.* [46]. All automatic detections were verified by a trained analyst (J.S.T.). A sum over all minutes with detections per day was computed. These daily sums were linearly adjusted, dividing by the percentage of effort per day. Weekly averages of these daily minutes with detections were calculated.

### (c) MFAS detection and characterization

The acoustic recordings were downsampled to a sampling rate of 10 kHz, and two analysts (A.E.S. and R.H.), trained to recognize MFAS signals, scanned long-term spectral averages (LTSAs) [28] over a frequency range of 10–5000 Hz to identify time periods with MFAS present. A 'packet' was defined as a tightly spaced cluster of pulses or pings, which occurred within a 1 kHz band between 2.5 and 4.5 kHz, with a pause between signals of no more than 0.1 s (electronic supplementary material, figure S1). To inspect the packets more closely, the analyst scanned spectrograms (Hann window, DFT = 1000, 50% overlap) in a 20 s window to log the start time, and the lowest and highest frequencies of packet components. When the start time of one packet occurred within 5 s of the start of a previous packet, they were combined into a single packet (electronic supplementary material, figure S1).

The acoustic energy of the MFAS was characterized based on MFAS packets. To minimize low-frequency ambient noise and focus on the energy band of MFAS, the data was filtered with a 10-pole Butterworth bandpass filter (2–4.95 kHz). The duration of the sonar packet was defined as the interval over which 90% of the sound energy arrived at the receiver, with the start and endpoints of an event at the 5% and 95% levels of cumulative energy within a time window [5]. A 10 s or 3 s window was used for sonar packets with multiple or single MFAS signals, respectively. Details of the signal level calculations are included in the electronic supplementary material.

The ability of an analyst to detect MFAS events depends on the received level of the signal and the underlying noise conditions. Based on the distribution of the received levels detected (electronic supplementary material, figure S2), a threshold of 115 dB re: 1 μPa was established such that signal characteristics were only reported for packets with a received level greater than the threshold (table 2). An additional subset of signals was removed from the analysis due to poor data quality. A portion of the low-frequency data (less than 5 kHz) collected in 2013 from the East HARP was not usable for detection of MFAS signals due to a hardware failure. This hardware failure did not affect the detectability of the higher frequency beaked whale signals. Due to limitations on analyst time, sonar packets were only analysed in the first 6 min at the beginning and middle of each hour (e.g. 12.00–12.06 and 12.30–12.36) during September 2014.

To obtain a record of naval ASW within the MIRC range, openly available sources were reviewed, including US Navy Press releases, newspaper reports and public internet news sources. This list is biased toward US naval activity, although other nations were involved in many training exercises, both with and without the US Navy.

## Note added in proof

After the manuscript was accepted for publication additional information was made available to the authors by the US Navy. Although the January 2019 beaked whale stranding occurred within the publicly reported dates for Exercise Sea Dragon (14–26 January 2019), the US Navy confirmed that there was no sonar usage associated with this training exercise, or elsewhere within the Mariana Islands Training and Testing area in the 6 days prior to the stranding. If this event is removed from the statistical analysis, there is a 1% probability (see electronic supplementary material, table S2) that three of eight beaked whale strandings occurred within 6 days after MFAS operations by chance. As discussed within the manuscript, the statistical analysis was limited to assessing the overlap between beaked whale strandings and known MFAS events (either via public reporting or through detection on passive acoustic devices—see figure 3). The Navy is working with NOAA to make the broader dataset, which is classified, available for further statistical analysis.

Data accessibility. Datasets containing detection times of beaked whales and sonar, along with the source code used to analyse the relationship of beaked whale strandings and Navy operations are uploaded to the Dryad Digital Repository: https://dx.doi.org/10.5061/dryad.7wm37pvnp [48]. Acoustic recordings containing the detections of beaked whales and MFAS are available from the East HARP (3 July 2012 to 12 May 2013) and West HARP (25 June 2012 to 4 March 2013). The raw acoustic data for this study are only partially available through Dryad because the volume of the entire dataset (greater than 16 TB) exceeds the Dryad data limitations (10 GB) at the time of publication. To obtain a copy of the raw acoustic data, please contact Erin Oleson (erin.oleson@noaa.gov) at NOAA's Pacific Islands Fisheries Science Center.

Authors' contributions. A.E.S. participated in the study design, analysed and interpreted the acoustic data, as well as drafted and revised the manuscript. R.L.B. collected and interpreted the stranding and naval operations data, and helped to draft and revise the manuscript. R.H. participated in the analysis and interpretation of acoustic data and revised the manuscript. J.S.T. analysed the acoustic beaked whale data and revised the manuscript. B.J.T. designed the methods for sonar analysis and revised the manuscript. E.M.O. participated in the study design, collected the acoustic data and revised the manuscript. S.B.-P. participated in the study design, and helped draft and revise the manuscript. All authors gave final approval for publication and agree to be held accountable for the work performed therein.

Competing interests. Several of the co-authors receive funding from the US Navy for other work unrelated to this study.

Funding. Funding for deployments and recoveries of HARPs and analysis of MFAS was provided by NOAA PIFSC. Funds for the detection and classification of beaked whale signals within the HARP datasets were provided by US Navy Pacific Fleet through interagency agreements (grant nos PIC-14-002, PIC-15-007, PIC-16-007), and the Cooperative Institute for Marine Ecosystems and Climate (grant no. NA15OAR4320071). This work was also supported by the NOAA John H. Prescott Marine Mammal Rescue Assistance Grant Program.

Acknowledgments. The authors would like to acknowledge the technicians and lab and field assistants at PIFSC and SIO for assistance with HARP preparation, deployment and recoveries. HARPs were deployed and recovered from the NOAA research vessels R/V Oscar Elton Sette and Hi'ialakai, and from the chartered vessel Sea Hunter. We gratefully acknowledge the assistance of Ben Sablan on Saipan for his efforts towards HARP preparation, transport, deployment, recovery and procurement of weight and other necessary supplies in this remote region. NOAA staff in Saipan—Mike Trianni, Dana Okano and Steve McKagan—have provided much needed assistance locating supplies, shipping and receiving HARP gear, and storing gear between deployments. We are very grateful for critical information and discussions with Kristi West regarding the beaked whale strandings included in this study. We also would like to thank Brent Tibbatts and the Guam Department of Agriculture, Division of Aquatic and Wildlife Resources for information on stranding records and reports. We also thank Jim Caretta for guidance on the statistical analysis. The manuscript was improved through the reviews provided by two anonymous reviewers, as well as the US Navy including Chip Johnson, Danielle Kitchen, Julie Rivers and Robert Uyeyama.

## Endnote

[1]Reported as about 12 ft (366 cm) but specimen looks smaller in photos.

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
