## [Reviewer comments · Proceedings of the Royal Society B: Biological Sciences]

Review History

Decision letter (RSPB-2020-0070.R0)

30-Jan-2020

Dear Dr Simonis

I am pleased to inform you that the Editor and Associate Editor are happy to publish the note with your paper. I'm afraid that you will need to submit your paper again as we need your paper to be submitted as a revision in order to pass it on to our production team successfully.

To upload your manuscript, log into <http://mc.manuscriptcentral.com/prsb> and enter your Author Centre, where you will find your manuscript title listed under "Manuscripts with Decisions." Under "Actions," click on "Create a Revision." Your manuscript number has been appended to denote a revision.

You will be unable to make your revisions on the originally submitted version of the manuscript. Instead, upload a new version through your Author Centre.

1) A text file of the manuscript (doc, txt, rtf or tex), including the references, tables (including captions) and figure captions. Please remove any tracked changes from the text before submission. PDF files are not an accepted format for the "Main Document".

2) A separate electronic file of each figure (tiff, EPS or print-quality PDF preferred). The format should be produced directly from original creation package, or original software format. Please note that PowerPoint files are not accepted.

3) Electronic supplementary material: this should be contained in a separate file from the main text and the file name should contain the author's name and journal name, e.g. authorname_procb_ESM_figures.pdf

All supplementary materials accompanying an accepted article will be treated as in their final form. They will be published alongside the paper on the journal website and posted on the online figshare repository. Files on figshare will be made available approximately one week before the accompanying article so that the supplementary material can be attributed a unique DOI. Please see: <https://royalsociety.org/journals/authors/author-guidelines/>

4) Data-Sharing and data citation

It is a condition of publication that data supporting your paper are made available. Data should be made available either in the electronic supplementary material or through an appropriate repository. Details of how to access data should be included in your paper. Please see <https://royalsociety.org/journals/ethics-policies/data-sharing-mining/> for more details.

<http://datadryad.org/submit?journalID=RSPB&manu=RSPB-2020-0070> which will take you to your unique entry in the Dryad repository.

Once again, thank you for submitting your manuscript to Proceedings B and I look forward to receiving your final version. If you have any questions at all, please do not hesitate to get in touch.

Sincerely,
Proceedings B Team
<mailto:proceedingsb@royalsociety.org>

Decision letter (RSPB-2020-0070.R1)

31-Jan-2020

Dear Dr Simonis

I am pleased to inform you that your manuscript entitled "Co-occurrence of beaked whale strandings and naval sonar in the Mariana Islands, Western Pacific" has been accepted for publication in Proceedings B.

Your article has been estimated as being 9 pages long. Our Production Office will be able to confirm the exact length at proof stage.

Open Access

Paper charges

Sincerely,

Proceedings B

Co-occurrence of beaked whale strandings and naval sonar in the Mariana Islands, Western Pacific

Anne E. Simonis, Robert L. Brownell, Bruce J. Thayre, Jennifer S. Trickey, Erin M. Oleson, Roderick Huntington and Simone Baumann-Pickering

Article citation details

Proc. R. Soc. B **286**: 20191500.
<http://dx.doi.org/10.1098/rspb.2019.1500>

Review timeline

Original submission: 29 July 2019
1st revised submission: 22 October 2019
2nd revised submission: 25 November 2019
Final acceptance: 25 November 2019

Note: Reports are unedited and appear as submitted by the referee. The review history appears in chronological order.

Review History

RSPB-2019-1500.R0 (Original submission)

Review form: Reviewer 1

Recommendation

Accept with minor revision (please list in comments)

Scientific importance: Is the manuscript an original and important contribution to its field?

Excellent

General interest: Is the paper of sufficient general interest?

Excellent

Quality of the paper: Is the overall quality of the paper suitable?

Good

Is the length of the paper justified?

Yes

Should the paper be seen by a specialist statistical reviewer?

No

Do you have any concerns about statistical analyses in this paper? If so, please specify them explicitly in your report.

No

It is a condition of publication that authors make their supporting data, code and materials available - either as supplementary material or hosted in an external repository. Please rate, if applicable, the supporting data on the following criteria.

Is it accessible?

Yes

Is it clear?

Yes

Is it adequate?

Yes

Do you have any ethical concerns with this paper?

No

Comments to the Author

The effects of anthropogenic sound on marine life is an important issue, and often critical information is lacking to assess or understand such effects. This study uses acoustic detections of both beaked whales and Navy sonar, combined with stranding records and available records on Navy exercises, and shows that Navy sonar is likely responsible for half the beaked whale strandings in the Mariana archipelago. The study also shows that strandings of single individuals can result from sonar use, which have previously not been attributed to Navy sonar. The approach is robust and the results are convincing, and this paper could be published with just a few minor edits. The title could be revised to greater emphasize the findings of beaked whale strandings in response to naval sonar use.

Minor comments

L43. "(MFA) sonar" should be changed to "sonar (MFAS)" to be consistent with how it is used throughout

L80-81. I presume the "Range Complex" is an area where the Navy operates, so it is not the activities of the Range Complex, it is the activities of the Navy, within the range complex.

L84. Should be "unidentified beaked whale (likely the ginkgo-toothed beaked whale, M. ginkgodens, characterized as "BWC" by [16])

L89. ASW is not defined anywhere?

L98. Already noted that BWC is likely the ginkgo-toothed beaked whale (note "beaked" in the name), so should change this to "...Blainville's, Cuvier's, and likely the ginkgo-toothed beaked whale."

L134. Should insert "from four of eight events", after the six of the ten Cuvier's beaked whales.

L151-152. This is the third time the connection between BWC and ginkgo-toothed beaked whale has been made – unnecessary repetition.

L175. “further” should be “farther from”

Table 1. Why is the species of the second record listed as “NA”, when the skull was collected and is at the Smithsonian? Presumably the species is known? There are nine strandings listed in the table but eight noted throughout the text. Given that the two from August 2011 are considered a single event throughout the text, some note to the table should be added to state this.

Table 2. Provide units for “packet duration”. Provide an explanation for SEL.

References. There are a lot of differences in formatting among the references, and some where the title (5, 11) or author initials (4) appear to be incorrect. These are just a few that stood out – they should all be gone through carefully and compared to the original sources.

Review form: Reviewer 2

Recommendation

Accept with minor revision (please list in comments)

Scientific importance: Is the manuscript an original and important contribution to its field?

Good

General interest: Is the paper of sufficient general interest?

Excellent

Quality of the paper: Is the overall quality of the paper suitable?

Good

Is the length of the paper justified?

Yes

Should the paper be seen by a specialist statistical reviewer?

No

Do you have any concerns about statistical analyses in this paper? If so, please specify them explicitly in your report.

Yes

It is a condition of publication that authors make their supporting data, code and materials available - either as supplementary material or hosted in an external repository. Please rate, if applicable, the supporting data on the following criteria.

Is it accessible?

Yes

Is it clear?

Yes

Is it adequate?

Yes

Do you have any ethical concerns with this paper?

No

Comments to the Author

This paper is particularly important because it describes data from the western Pacific (which has had little coverage for this work) and is the first study I am aware of that was able to compare recordings of MFAS and beaked whales outside of an instrumented Navy range and correlate documented ASW sonar transmissions with strandings. This means that it would be particularly important and useful for the paper to compare how often MFAS was recorded against the publicly available reports of sonar usage. I strongly urge the authors to include this result, which will probably require a more intensive analysis than presented here. Duration of ASW exercises should be compared to the duty cycling of the recordings, which is not described in enough detail, to estimate $p(\text{detection})$ of exercises from the duty-cycled data. Analysis in abstract and body of paper is a bit jumbled between the longer period associating reports of exercises and strandings vs the shorter period of acoustic data. If paper keeps both, better to separate more clearly. In addition, many papers have reported silencing of beaked whales during and after exercises. This paper should analyse a similar effect for before/ during/ after MFAS sessions, perhaps as $f(\text{RL})$. Discussion should include a section on how to design studies using passive acoustic monitoring to test for link between beaked whale presence, sonar, and stranding. If length is a concern, discussion could reduce poorly founded speculation about whether risk is higher in Marianas and if so, what might cause that.

L54-57 – give a citation for “the vast majority of 55 atypical mass beaked whale stranding events that occurred during the 1960s and 1970s in Japan, 56 southern California,”

Line 68 change “first beaked whale stranding on Guam” to “first beaked whale stranding recorded on Guam”

Lines 113-117 change “When MFAS packets were 114 detected, they generally occurred in consecutive bouts with fewer than two minutes between 115 sonar packets, followed by “breaks” in sonar activity, which ranged from 15 minutes to nearly 3 116 hours (Table 2). MFAS events were detected on a total of 35 days between 2011 and 2014, with 117 MFAS events lasting from 1 to 18 days (Table 2).”

To

“MFAS events were detected on a total of 35 days between 2011 and 2014, with 117 MFAS events lasting from 1 to 18 days (Table 2). When MFAS packets were 114 detected, they generally occurred in consecutive bouts with fewer than two minutes between 115 sonar packets, followed by “breaks” in sonar activity, which ranged from 15 minutes to nearly 3 116 hours (Table 2).”

Lines 136-146 The statistical test assumed an equal probability of beaked whale availability for stranding thought the time period, but the acoustic detections show otherwise. Unless authors

can more convincingly justify assumption of equal probability, better to draw the null hypothesis dates from a distribution that more closely matches the detections of each species.

Lines 157-159 another indicator of low prob of detecting beaked whales when they are in the area is the low number of minutes/day during days when they ARE detected.

Sentence in 200-203 does not make sense. Clarify.

Sentence in 215-218 does not follow. Risk of sonar may be just as high elsewhere.

Line 231 – beaked whale move off range during and after exercise but it is not known if they abandon foraging areas or just shift foraging areas

Figure 2 needs to label MFA in bottom cell

Decision letter (RSPB-2019-1500.R0)

23-Sep-2019

Dear Dr Simonis:

Your manuscript has now been peer reviewed and the reviews have been assessed by an Associate Editor. The reviewers' comments (not including confidential comments to the Editor) and the comments from the Associate Editor are included at the end of this email for your reference. As you will see, the reviewers and the Editors have raised some concerns with your manuscript and we would like to invite you to revise your manuscript to address them.

Research ethics:

Use of animals and field studies:

If you wish to submit your data to Dryad (<http://datadryad.org/>) and have not already done so you can submit your data via this link [http://datadryad.org/submit?journalID=RSPB&manu=\(Document not available\)](http://datadryad.org/submit?journalID=RSPB&manu=(Document%20not%20available)), which will take you to your unique entry in the Dryad repository.

Please submit a copy of your revised paper within three weeks. If we do not hear from you within this time your manuscript will be rejected. If you are unable to meet this deadline please let us know as soon as possible, as we may be able to grant a short extension.

Best wishes,
Dr Sasha Dall
mailto: proceedingsb@royalsociety.org

Associate Editor
Board Member: 1

Comments to Author:

Both reviewers have identified that this manuscript would be publishable following revision which I agree with. Please consider the revisions being asked for carefully and provide a detailed response against each.

I have a number of editorial suggestions for tidying up the content of the manuscript to add:

Referencing to "Guam and the CNMI", "the Mariana archipelago", "the northern Mariana Islands", "the Mariana Islands", "Guam and the Mariana Islands", "Guam and the Northern Mariana Islands" are all used interchangeably throughout the manuscript which can be confusing to the reader. Referencing to these islands should be consistent throughout the manuscript and it should be clear what you are referring to - e.g. the Mariana Archipelago consists of the islands of Guam to the south and the northern Mariana Islands (including Saipan and Tinian Islands) to the north.

Similarly "MFAS", "MFAS events" and "MFAS operations" are used interchangeably. Please be consistent in what is being referred to. Associated, the term "use of ASW" is used to refer directly to MFAS which is not entirely appropriate because ASW may not include the use of MFAS 100% of the time.

Line 26: should be amended to "...in the waters offshore of the Mariana Islands..." I don't think the navy uses ASW in the Mariana Islands themselves

Line 27 : should be amended to "...off Saipan and Tinian Islands..." see also line 32

Line 43: refers to MFA and then line 44 refers to MFAS (which is then used throughout the manuscript). Choose one and be consistent in use.

Lines 53-57 needs some further context - what does "the vast majority" refer to? This sentence seems speculative at best as currently presented

Line 60: location should be plural given you are referring to Guam and the CNMI

Line 80: I think this should be "the activities of the US Navy in the Mariana Island Range Complex"

Lines 81-85: this sentence currently reads as though visual surveys have detected echolocation clicks which I don't think is the intent

Line 87: amend to "...near Saipan and Tinian Islands in the northern Mariana Islands..."

Line 139: replace "strandings" with "stranding events"

Line 140: amend to "...number of simulated stranding events..."

Line 143: amend to "...naval event/stranding event..."

Line 166: amend to "...event on Saipan Island, a location where..."

Line 174: amend to "...(west coast of Saipan Island)..."

Line 182: note that no reference to canyons in the area of the study has been made prior to this - is the region one of a high density of canyons? Why might a high density of canyons be an important factor?

Line 186: stranding rates haven't increased since 2007 (at least according to Table 1) - they have consistently occurred at 2 per year. What has actually occurred is that beaked whales were not

recorded in the stranding record prior to 2007 and since 2007 have occurred at constant rates in the stranding record. Please amend the wording so that it directly reflects the data being presented.

Lines 190 and 191: amend to identify Saipan as Saipan Island

Lines 200-207: not clear what is being said here

Lines 208-210: the definition of a mass stranding is 2 or more animals, excepting mothers with dependent calves, so if the assessment is of mass strandings then it has to include strandings of 2 or more animals - this sentence as it is currently written doesn't make sense.

Lines 228-229 and lines 230-231 appear to be saying the same things which is then repeated in the context of the study area on lines 233-236 - please reduce the repetition or clarify the differences

Line 237: amend to "Saipan Island"

Lines 237-242: not clear what the context of this paragraph is or the point being made

Lines 252: delete "well" - they've been documented; "well" is a subjective term.

Line 252: amend to "Saipan Island"

Reviewer(s)' Comments to Author:

Referee: 1

Comments to the Author(s)

The effects of anthropogenic sound on marine life is an important issue, and often critical information is lacking to assess or understand such effects. This study uses acoustic detections of both beaked whales and Navy sonar, combined with stranding records and available records on Navy exercises, and shows that Navy sonar is likely responsible for half the beaked whale strandings in the Mariana archipelago. The study also shows that strandings of single individuals can result from sonar use, which have previously not been attributed to Navy sonar. The approach is robust and the results are convincing, and this paper could be published with just a few minor edits. The title could be revised to greater emphasize the findings of beaked whale strandings in response to naval sonar use.

Minor comments

L43. "(MFA) sonar" should be changed to "sonar (MFAS)" to be consistent with how it is used throughout

L80-81. I presume the "Range Complex" is an area where the Navy operates, so it is not the activities of the Range Complex, it is the activities of the Navy, within the range complex.

L84. Should be "unidentified beaked whale (likely the ginkgo-toothed beaked whale, M. ginkgodens, characterized as "BWC" by [16])

L89. ASW is not defined anywhere?

L98. Already noted that BWC is likely the ginkgo-toothed beaked whale (note "beaked" in the name), so should change this to "...Blainville's, Cuvier's, and likely the ginkgo-toothed beaked whale."

L134. Should insert "from four of eight events", after the six of the ten Cuvier's beaked whales.

L151-152. This is the third time the connection between BWC and ginkgo-toothed beaked whale has been made - unnecessary repetition.

L175. “further” should be “farther from”

Table 1. Why is the species of the second record listed as “NA”, when the skull was collected and is at the Smithsonian? Presumably the species is known? There are nine strandings listed in the table but eight noted throughout the text. Given that the two from August 2011 are considered a single event throughout the text, some note to the table should be added to state this.

Table 2. Provide units for “packet duration”. Provide an explanation for SEL.

References. There are a lot of differences in formatting among the references, and some where the title (5, 11) or author initials (4) appear to be incorrect. These are just a few that stood out – they should all be gone through carefully and compared to the original sources.

Referee: 2

Comments to the Author(s)

This paper is particularly important because it describes data from the western Pacific (which has had little coverage for this work) and is the first study I am aware of that was able to compare recordings of MFAS and beaked whales outside of an instrumented Navy range and correlate documented ASW sonar transmissions with strandings. This means that it would be particularly important and useful for the paper to compare how often MFAS was recorded against the publicly available reports of sonar usage. I strongly urge the authors to include this result, which will probably require a more intensive analysis than presented here. Duration of ASW exercises should be compared to the duty cycling of the recordings, which is not described in enough detail, to estimate $p(\text{detection})$ of exercises from the duty-cycled data. Analysis in abstract and body of paper is a bit jumbled between the longer period associating reports of exercises and strandings vs the shorter period of acoustic data. If paper keeps both, better to separate more clearly. In addition, many papers have reported silencing of beaked whales during and after exercises. This paper should analyse a similar effect for before/during/after MFAS sessions, perhaps as $f(\text{RL})$. Discussion should include a section on how to design studies using passive acoustic monitoring to test for link between beaked whale presence, sonar, and stranding. If length is a concern, discussion could reduce poorly founded speculation about whether risk is higher in Marianas and if so, what might cause that.

L54-57 – give a citation for “the vast majority of 55 atypical mass beaked whale stranding events that occurred during the 1960s and 1970s in Japan, 56 southern California,”

Line 68 change “first beaked whale stranding on Guam” to “first beaked whale stranding recorded on Guam”

Lines 113-117 change “When MFAS packets were 114 detected, they generally occurred in consecutive bouts with fewer than two minutes between 115 sonar packets, followed by “breaks” in sonar activity, which ranged from 15 minutes to nearly 3 116 hours (Table 2). MFAS events were detected on a total of 35 days between 2011 and 2014, with 117 MFAS events lasting from 1 to 18 days (Table 2).”

To

“MFAS events were detected on a total of 35 days between 2011 and 2014, with 117 MFAS events lasting from 1 to 18 days (Table 2). When MFAS packets were 114 detected, they generally occurred in consecutive bouts with fewer than two minutes between 115 sonar packets, followed by “breaks” in sonar activity, which ranged from 15 minutes to nearly 3 116 hours (Table 2).”

Lines 136-146 The statistical test assumed an equal probability of beaked whale availability for stranding thought the time period, but the acoustic detections show otherwise. Unless authors can more convincingly justify assumption of equal probability, better to draw the null hypothesis dates from a distribution that more closely matches the detections of each species.

Lines 157-159 another indicator of low prob of detecting beaked whales when they are in the area is the low number of minutes/day during days when they ARE detected.

Sentence in 200-203 does not make sense. Clarify.

Sentence in 215-218 does not follow. Risk of sonar may be just as high elsewhere.

Line 231 – beaked whale move off range during and after exercise but it is not known if they abandon foraging areas or just shift foraging areas

Figure 2 needs to label MFA in bottom cell

Author's Response to Decision Letter for (RSPB-2019-1500.R0)

See Appendix A.

Decision letter (RSPB-2019-1500.R1)

01-Nov-2019

Dear Dr Simonis

I am pleased to inform you that your Review manuscript RSPB-2019-1500.R1 entitled "Co-occurrence of beaked whale strandings and naval sonar in the Mariana Islands, Western Pacific" has been accepted for publication in Proceedings B.

The referee(s) do not recommend any further changes. Therefore, please proof-read your manuscript carefully and upload your final files for publication. Because the schedule for publication is very tight, it is a condition of publication that you submit the revised version of your manuscript within 7 days. If you do not think you will be able to meet this date please let me know immediately.

To upload your manuscript, log into <http://mc.manuscriptcentral.com/prsb> and enter your Author Centre, where you will find your manuscript title listed under "Manuscripts with Decisions." Under "Actions," click on "Create a Revision." Your manuscript number has been appended to denote a revision.

You will be unable to make your revisions on the originally submitted version of the manuscript. Instead, upload a new version through your Author Centre.

1) A text file of the manuscript (doc, txt, rtf or tex), including the references, tables (including captions) and figure captions. Please remove any tracked changes from the text before submission. PDF files are not an accepted format for the "Main Document".

2) A separate electronic file of each figure (tiff, EPS or print-quality PDF preferred). The format should be produced directly from original creation package, or original software format. Please note that PowerPoint files are not accepted.

3) Electronic supplementary material: this should be contained in a separate file from the main text and the file name should contain the author's name and journal name, e.g. `authurname_procb_ESM_figures.pdf`

All supplementary materials accompanying an accepted article will be treated as in their final form. They will be published alongside the paper on the journal website and posted on the online figshare repository. Files on figshare will be made available approximately one week before the accompanying article so that the supplementary material can be attributed a unique DOI. Please see: <https://royalsociety.org/journals/authors/author-guidelines/>

4) Data-Sharing and data citation

It is a condition of publication that data supporting your paper are made available. Data should be made available either in the electronic supplementary material or through an appropriate repository. Details of how to access data should be included in your paper. Please see <https://royalsociety.org/journals/ethics-policies/data-sharing-mining/> for more details.

<http://datadryad.org/submit?journalID=RSPB&manu=RSPB-2019-1500.R1> which will take you to your unique entry in the Dryad repository.

Once again, thank you for submitting your manuscript to Proceedings B and I look forward to receiving your final version. If you have any questions at all, please do not hesitate to get in touch.

Sincerely,
Dr Sasha Dall
Editor, Proceedings B
<mailto:proceedingsb@royalsociety.org>

Associate Editor
Comments to Author:

There are still some inconsistencies in text and some clarification of text required as well as some speculative statements that need to be addressed. A marked up version of the manuscript identifying these revisions is attached.

Decision letter (RSPB-2019-1500.R2)

25-Nov-2019

Dear Dr Simonis

I am pleased to inform you that your manuscript entitled "Co-occurrence of beaked whale strandings and naval sonar in the Mariana Islands, Western Pacific" has been accepted for publication in Proceedings B.

Your article has been estimated as being 9 pages long. Our Production Office will be able to confirm the exact length at proof stage.

Open Access

Paper charges

You are allowed to post any version of your manuscript on a personal website, repository or preprint server. However, the work remains under media embargo and you should not discuss it

with the press until the date of publication. Please visit <https://royalsociety.org/journals/ethics-policies/media-embargo> for more information.

Sincerely,
Proceedings B
<mailto:proceedingsb@royalsociety.org>

Appendix A

The authors response to referees is shown in blue font following each point. Referenced line numbers for the revised word document refer to line numbers viewed with tracked changes in the 'All Markup' mode.

Response to Referees:

Associate Editor

Referencing to "Guam and the CNMI", "the Mariana archipelago", "the northern Mariana Islands", "the Mariana Islands", "Guam and the Mariana Islands", "Guam and the Northern Mariana Islands" are all used interchangeably throughout the manuscript which can be confusing to the reader. Referencing to these islands should be consistent throughout the manuscript and it should be clear what you are referring to - e.g. the Mariana Archipelago consists of the islands of Guam to the south and the northern Mariana Islands (including Saipan and Tinian Islands) to the north.

We now define the Mariana Archipelago in the introduction (line 63): "The Mariana Archipelago, consisting of the islands of Guam to the south and the Commonwealth of the Northern Mariana Islands (including Saipan and Tinian, hereafter referred to as the Northern Mariana Islands) to the north,..."

Subsequent references have been changed to refer to individual islands, "Northern Mariana Islands", or the Mariana Archipelago.

Similarly "MFAS", "MFAS events" and "MFAS operations" are used interchangeably. Please be consistent in what is being referred to. Associated, the term "use of ASW" is used to refer directly to MFAS which is not entirely appropriate because ASW may not include the use of MFAS 100% of the time.

All references to MFA/MFAS/MFAS operations have been changed to "MFAS". We made multiple clarifications in the text to refer to ASW for reported anti-submarine operations, and only use the term "MFAS" when MFAS was detected in the acoustic record, or reported from the literature.

Line 35: Changed MFA to ASW

Lines 90-91: Changed 'multinational naval activities' to "multinational ASW activities" and deleted "known or assumed to include ASW"

Line 124: replaced "naval" with "ASW", and deleted "that utilize MFAS"

Line 134: added "ASW" to "naval ASW exercise"

Line 187: inserted "ASW" to "multinational naval ASW operations", and deleted "that utilize MFAS"

Line 246: inserted text in quotes: The U.S. Navy has also confirmed that "the MFAS used in" major multinational naval "ASW" training exercises was associated with the beaked whale stranding events...

Line 214: replace "military" with "ASW"

Line 215: replace "military" with "ASW"

Line 253: replace "military" with "ASW"

Line 253: replace "military" with "ASW"

Line 256: replace "military" with "ASW"

Line 320: replace "MFAS" with "ASW"

Line 26: should be amended to "...in the waters offshore of the Mariana Islands..." I don't think the navy uses ASW in the Mariana Islands themselves

Done.

Line 27 : should be amended to "...off Saipan and Tinian Islands..." see also line 32

Changed to "...near the islands of Saipan and Tinian", and "the Mariana Archipelago" in line 32
Similar to how it would be incorrect to refer to "Guam Island" or "O'ahu Island", the word "Island" is not in the official name of Saipan or Tinian. Readers unfamiliar with the geographic area are made aware that Saipan and Tinian are islands from the definition of the Mariana Archipelago in the intro (lines 63-65)

Line 43: refers to MFA and then line 44 refers to MFAS (which is then used throughout the manuscript). Choose one and be consistent in use.

All references have been changed to "MFAS"

Lines 53-57 needs some further context - what does "the vast majority" refer to? This sentence seems speculative at best as currently presented

The term "vast majority" has been removed, as well as the reference to "southern California". A citation has been added to support the statement about strandings in Japan.

Line 60: location should be plural given you are referring to Guam and the CNMI

Location is singular as it now refers to the Archipelago

Line 80: I think this should be "the activities of the US Navy in the Mariana Island Range Complex"

Correct. Text changed to: "...the activities of the U.S. Navy in the Mariana Island Range Complex"

Lines 81-85: this sentence currently reads as though visual surveys have detected echolocation clicks which I don't think is the intent

Correct. Divided into two sentences. changed to "Visual surveys since 2007 have documented Cuvier's, Blainville's (*Mesoplodon densirostris*) and unconfirmed *Mesoplodon* spp. beaked whales in deep waters (>650 m) (Fulling et al. 2011). Since 2010, acoustic monitoring has documented echolocation clicks from Cuvier's, Blainville's and an unidentified (likely the ginkgo-toothed whale, *M. ginkgodens*) "BWC" beaked whale (Baumann-Pickering et al. 2014) near Saipan and Tinian throughout the year (Oleson et al. 2015; Au and Lammers 2016; Klinck et al. 2015, 2016; Munger et al. 2015)."

Line 87: amend to "...near Saipan and Tinian Islands in the northern Mariana Islands..."

Removed "in the Mariana Islands". Text now reads "The purpose of this study was to document the seasonal acoustic presence of beaked whales near Saipan and Tinian using High-frequency Acoustic Recording Packages"

Line 139: replace "strandings" with "stranding events"

Done

Line 140: amend to "...number of simulated stranding events..."

Done

Line 143: amend to "...naval event/stranding event..."

Done

Line 166: amend to "...event on Saipan Island, a location where..."

See earlier response as to why "Saipan Island" is not appropriate

Changed to "on Saipan, a location where ..."

Line 174: amend to "... (west coast of Saipan Island)..."

See earlier response as to why "Saipan Island" is not appropriate

Line 182: note that no reference to canyons in the area of the study has been made prior to this - is the region one of a high density of canyons? Why might a high density of canyons be an important factor? Submarine canyons are often indicative of beaked whale habitat, so we replaced the text "in an area with submarine canyons", with "in beaked whale habitat" for clarity.

Line 186: stranding rates haven't increased since 2007 (at least according to Table 1) - they have consistently occurred at 2 per year. What has actually occurred is that beaked whales were not recorded in the stranding record prior to 2007 and since 2007 have occurred at constant rates in the stranding record. Please amend the wording so that it directly reflects the data being presented.

Section heading changed to "Association of beaked whale stranding events with MFAS in the Mariana Archipelago"

It is not accurate to say that the stranding rate has been "constant", given occasional breaks of 2 or 3 years in the stranding record.

Line 186 changed to: Since 2007 there has been a strong association between beaked whale stranding events with the presence of multinational naval training operations that utilize MFAS

Lines 190 and 191: amend to identify Saipan as Saipan Island

No change made. See above comment re: "Saipan Island"

Lines 200-207: not clear what is being said here

The authors wished to retain this paragraph because it supports the lack of reports of beaked whale stranding events before 2007 as an indicator of no or very few beaked whale stranding events.

However, given the page limits of the journal, this section has been removed for brevity.

Lines 208-210: the definition of a mass stranding is 2 or more animals, excepting mothers with dependent calves, so if the assessment is of mass strandings then it has to include strandings of 2 or more animals - this sentence as it is currently written doesn't make sense.

Adjusted text for clarity: "Previous studies suggest that 9% of global beaked whale mass strandings are associated with naval operations involving MFAS (D'Amico et al. 2009); however, by only considering

mass strandings (2 or more animals, excepting mothers with dependent calves), this is a conservative metric because single animal strandings may also be associated with MFAS.”

Lines 228-229 and lines 230-231 appear to be saying the same things which is then repeated in the context of the study area on lines 233-236 - please reduce the repetition or clarify the differences

Reduced and clarified these two points. The text now reads

“Especially in a pristine acoustic environment, beaked whales have shown strong avoidance responses to both near and distant MFAS (Wensveen et al. 2019). Conversely, after decades of exposure to MFAS disturbances, some resident beaked whales near navy ranges may habituate to sonar or learn to abandon preferred habitat during MFAS operations (McCarthy et al. 2011; Moretti et al. 2014; Tyack et al. 2011); however, there may still be high energetic costs associated with avoiding MFAS (Southall et al. 2019).”

Line 237: amend to "Saipan Island"

No change made. See above comment re: "Saipan Island"

Lines 237-242: not clear what the context of this paragraph is or the point being made

The presence of nematodes has been suggested as a potential factor leading to the stranding of beaked whales in Guam and Saipan in public news reports, as well as the Navy’s Environmental Impact Statement. We strongly feel that it is important to report that these nematodes are very common in both healthy and stranded animals and that it is unlikely that the nematodes were an important factor in the sonar-associated strandings in the Mariana Archipelago. Due to strict page limits with the journal, we omitted one additional reference that documents *Crassicauda* in Cuvier’s beaked whales: (Komnenou et al. 2017).

We also added text to the paragraph to justify the discussion of *Crassicauda* including:

Lines 330-359: “When one of the two 2011 Saipan beaked whales was examined, we (KLW) found a heavy infestation of giant nematodes (*Crassicauda* sp.) in both kidneys of the 4.39 m male specimen. The U.S. Navy 2019 EIS for the Mariana Islands Range Complex suggested that this heavy parasite load could be a potential factor leading to the stranding, because the whale was already compromised (Naval Facilities Engineering Command Pacific 2019). However, we consider this unlikely, because these nematodes are observed in most beaked whales that have been examined, regardless of the cause of death. They are usually found in healthy beaked whales taken by Japanese whalers (Brownell, published data) and stranded beaked whales (single and mass strandings) of various species, including Cuvier’s beaked whales (Bernaldo de Quirós et al. 2019; Díaz-Delgado et al. 2016; Tajima et al. 2015). Therefore, we believe the strong relationship between Cuvier’s beaked whale stranding events and MFAS reported here suggest that MFAS, and not these commonly occurring parasites, was the primary factor causing half of the beaked whale strandings in the Mariana Archipelago since 2007.”

Lines 252: delete "well" - they've been documented; "well" is a subjective term.

Done

Line 252: amend to "Saipan Island"

No change made. See above comment re: "Saipan Island"

Reviewer(s)' Comments to Author:

Referee: 1

Comments to the Author(s)

The effects of anthropogenic sound on marine life is an important issue, and often critical information is lacking to assess or understand such effects. This study uses acoustic detections of both beaked whales and Navy sonar, combined with stranding records and available records on Navy exercises, and shows that Navy sonar is likely responsible for half the beaked whale strandings in the Mariana archipelago. The study also shows that strandings of single individuals can result from sonar use, which have previously not been attributed to Navy sonar. The approach is robust and the results are convincing, and this paper could be published with just a few minor edits. The title could be revised to greater emphasize the findings of beaked whale strandings in response to naval sonar use.

Minor comments

L43. "(MFA) sonar" should be changed to "sonar (MFAS)" to be consistent with how it is used throughout

Done

L80-81. I presume the "Range Complex" is an area where the Navy operates, so it is not the activities of the Range Complex, it is the activities of the Navy, within the range complex.

Text changed to: "...impacted by military activities in the Mariana Island Range Complex"

L84. Should be "unidentified beaked whale (likely the ginkgo-toothed beaked whale, *M. ginkgodens*, characterized as "BWC" by [16])

Text changed to: "Since 2010, acoustic monitoring has documented echolocation clicks from Cuvier's, Blainville's, and an unidentified beaked whale (likely the ginkgo-toothed whale, *M. ginkgodens* characterized as "BWC" by (Baumann-Pickering et al. 2014)) near Saipan and Tinian throughout the year (Oleson et al. 2015; Au and Lammers 2016; Klinck et al. 2015, 2016; Munger et al. 2015).

L89. ASW is not defined anywhere?

The first mention of ASW in the abstract and main text (line 94) is now defined as 'antisubmarine warfare'.

L98. Already noted that BWC is likely the ginkgo-toothed beaked whale (note "beaked" in the name), so should change this to "...Blainville's, Cuvier's, and likely the ginkgo-toothed beaked whale."

Other authors have described a likely connection between the "BWC" signal and the ginkgo-toothed beaked whale based on geographic overlaps, however the acoustic signal has never been recorded in the known presence of a ginkgo-toothed beaked whale nor does there exist other evidence of the connection. Considering this, along with the consideration that there have been other recent discoveries

of new beaked whale species, we prefer to take a conservative approach and only report the known presence of the “BWC” signal.

The text has been changed to note the potential attribution of the “BWC” signal to the ginkgo-toothed beaked whale by other researchers once, and the remaining references to ginkgo-toothed beaked whales have been removed (Lines 98 and 151).

L134. Should insert “from four of eight events”, after the six of the ten Cuvier’s beaked whales.
Done

L151-152. This is the third time the connection between BWC and ginkgo-toothed beaked whale has been made – unnecessary repetition.
The reference to ginkgo-toothed beaked whale was removed.

L175. “further” should be “farther from”
Done

Table 1. Why is the species of the second record listed as “NA”, when the skull was collected and is at the Smithsonian? Presumably the species is known? There are nine strandings listed in the table but eight noted throughout the text. Given that the two from August 2011 are considered a single event throughout the text, some note to the table should be added to state this.

Yes! Good catch. The species of the 2008 stranding is known, and is Cuvier’s beaked whale. As of the time of this revision, the Smithsonian has not been able to locate the skull in their records, however photos of the stranded animal were reviewed to confirm the species, sex, and size. The table has been changed to reflect these details.

We also added to “Notes” column of the Table re: August 2011 strandings: “first day of stranding event” and “second day of stranding event” to clarify that these are from a single stranding event

Table 2. Provide units for “packet duration”. Provide an explanation for SEL.
Done

References. There are a lot of differences in formatting among the references, and some where the title (5, 11) or author initials (4) appear to be incorrect. These are just a few that stood out – they should all be gone through carefully and compared to the original sources.

Titles corrected in (5,11)

Author initials corrected in (4)

Added Institution for (14)

Added Book title (18, 32)

Referee: 2

Comments to the Author(s)

This paper is particularly important because it describes data from the western Pacific (which has had little coverage for this work) and is the first study I am aware of that was able to compare recordings of MFAS and beaked whales outside of an instrumented Navy range and correlate documented ASW sonar transmissions with strandings. This means that it would be particularly important and useful for the paper to compare how often MFAS was recorded against the publicly available reports of sonar usage. I strongly urge the authors to include this result, which will probably require a more intensive analysis than presented here. Duration of ASW exercises should be compared to the duty cycling of the recordings, which is not described in enough detail, to estimate $p(\text{detection})$ of exercises from the duty-cycled data.

We now include text describing how many publicly reported events were acoustically detected (Lines 167-171): “The U.S. Navy reported 4 major international antisubmarine operations while acoustic recordings were collected from at least one of two HARP locations. One of these events (Valiant Shield V: September 15-23 2014) was detected acoustically on September 15-16, 21-22, with MFAS also detected during 7 days prior and 5 days following the respective start and end dates of the operation. MFAS was also detected on 17 days that were not included in publicly reported events.”

The duration of MFAS encounters is now included in Table 2, and the duty cycle for each deployment is now shown under the bottom panel in Figures 1 and 2. The recording periods were always 5 minutes long, however we used a variety of duty cycles. The longest gap in active recording due to duty cycle was during 2010 at the West HARP, where we recorded 5 out of every 40 minutes. The duty cycle limited the detection of events with durations less than 35 minutes in 2010, 15 minutes in 2011, or 1-2 minutes in 2012-2014. The majority of MFAS encounters that we observed had durations longer than one hour, although they ranged from 1 minute to 18 hours 50 minutes. For ASW events spanning multiple days (true of all reported major multinational operations), the duty cycle of the recordings would likely not be a significant limiting factor for the detection probability.

Text has been added to better describe the impact of the duty cycle on detection probability: Lines 246-251: “The duty cycled recordings limited the detection of MFAS events with durations less than 35 minutes in 2010, 15 minutes in 2011, and 1-2 minutes in 2012-2014; however, the majority of MFAS encounters that we observed had durations longer than one hour, suggesting the duty cycle did not significantly limit MFAS detection overall. The duty-cycled nature of the recordings results in an incomplete report of acoustic activity at the recording locations; however, the signal characteristics reported here should be representative of the signals that were not recorded.”

In order to estimate the true probability of detection, we would need to know the position of the MFAS source, the source level, and the times that MFAS was in use. Unfortunately, the Navy does not report the precise times when MFAS is operational, nor the position of the MFAS sources during ASW operations, because that is considered classified knowledge. We do know that the source level of MFAS is estimated to be 217-235 dB (Commander United States Navy Pacific Fleet 2010). We estimated a potential detection area considering the expected absorption coefficient in the waters around the Mariana Archipelago (0.2 dB/km @3.5 kHz, 25C @ 10 m depth, 35 ppt salinity, pH 8) and a rough estimate of ambient noise based on sea state (@3.5 kHz, sea state 1=47 dB, sea state 2=53 dB, sea state 4=60 dB, sea state 6=66 dB) (Wenz 1962). We then divided the detection area in half to account for the steep local bathymetry or shadowing provided by the islands at each recording location. We estimate

Commented [AS1]: Add text to Figure captions during re-submission to reflect this

that the detection area for the two HARPs combined could range between 76,026 – 226,822 km², which is 4-12% of the total area (1,872,094 km²) of the Mariana Islands Range Complex (Naval Facilities Engineering Command Pacific 2019). Considering our limited detection area, and without knowledge of the position and timing of MFAS operations, our ability to estimate the true probability of detection of ASW events is severely limited.

Analysis in abstract and body of paper is a bit jumbled between the longer period associating reports of exercises and strandings vs the shorter period of acoustic data. If paper keeps both, better to separate more clearly.

We adjusted the text in the introduction, as well as the subheadings, to help clarify this distinction:

Line 108: We document acoustic activity of beaked whales and MFAS over 2010 – 2014, and also reviewed unclassified, publicly available reports of multinational naval activities known or assumed to include MFAS over the longer time period of 2006 - 2019.

Changed subheadings in the Results & Discussion sections to “Presence of military sonar (2010-2014)” and “Association of beaked whale stranding events with ASW training (2006 – 2019)”

In addition, many papers have reported silencing of beaked whales during and after exercises. This paper should analyse a similar effect for before/during/after MFAS sessions, perhaps as f(RL). We also were interested in looking into changes in beaked whale vocal behavior with MFAS activity; however, with so few MFAS encounters at the two locations from 2010 to 2014, the statistical significance was very low or unavailable for most tests. We believe that we don't have adequate statistical power to describe changes in beaked whale activity before, during, and after MFAS events. Summary tables of these analyses are now included with the supplemental materials.

Please note that data collection in this region is ongoing, and we are currently seeking funding to support an analysis of the expanded dataset to assess the behavioral response of beaked whales to MFAS using passive acoustic monitoring.

Wilcoxon Signed-Rank Tests:West HARP MFAS Encounters

1. August 21, 2011
2. August 28-September 1, 2012
3. September 21, 2012
4. December 16, 2013

Median Number Of Hours Present Per Day In The Week Prior To And After MFAS Encounter at the West HARP

Species	Before	During	After	p-value (Before-During)	p-value (During-After)
Cuvier's beaked whale	1	0	1	0.09	0.15
	1	0	1		
	2	0	0		
	1	0	1		
Blainville's beaked whale	1	0	1	0.17	0.17
	2	1	4		
	0	0	0		
	2	0	1		

Median Number of Hours Present In The Day Prior To And After MFAS Encounter at the West HARP

Species	Before	During	After	p-value (Before-During)	p-value (During-After)
Cuvier's beaked whale	1	0	0	1.0	NA
	0	0	0		
	0	0	0		
	0	0	0		
Blainville's beaked whale	1	0	0	1	NA
	0	1	1		
	0	0	0		
	0	0	0		

East HARP MFAS Encounters

1. August 21, 2011
2. August 27-September 9, 2012
3. September 21, 2012 (not used for weekly analysis)
4. January 16, 2014
5. September 8-17, 2014
6. September 21-28, 2014 (combined with Sept 8-17,2014 for weekly analysis)

Median Number Of Hours Present Per Day In The Week Prior To And After MFAS Encounter at the West HARP

Species	Before	During	After	p-value (Before-During)	p-value (During-After)
Cuvier's beaked whale	0 0 0 0	0 0 0 0	1 0 0 0	NA	1
Blainville's beaked whale	1 1 3 2	0 1 2 1	0 2 2 2	0.15	0.35

Median Number of Hours Present In The Day Prior To And After MFAS Encounter at the West HARP

Species	Before	During	After	p-value (Before-During)	p-value (During-After)
Cuvier's beaked whale	0 0 0 0 0 0	0 0 0 0 0 0	0 0 0 0 0 0	NA	NA
Blainville's beaked whale	0 1 0 5 0 0	0 1 0 2 1 1	0 0 0 0 0 0	1	0.09

Discussion should include a section on how to design studies using passive acoustic monitoring to test for link between beaked whale presence, sonar, and stranding. If length is a concern, discussion could reduce poorly founded speculation about whether risk is higher in Marianas and if so, what might cause that.

A paragraph has been added to the discussion including a few suggestions for future studies:
Line 353: “Looking into the future, optimal investigations of beaked whale behavior and MFAS using passive acoustic monitoring should incorporate a high density of acoustic sensors in a variety of habitats, capable of recording continuously over multiple seasonal cycles. Consistent stranding networks are needed to monitor and respond to individual and mass strandings in time to investigate the hypotheses associated with acoustic-barotrauma (deQuiros et al 2019). Ideally, the full disclosure of the timing and position of MFAS events would support more powerful conclusions about the behavioral response of animals and the potential risk for sonar-associated strandings. “

L54-57 – give a citation for “the vast majority of
55 atypical mass beaked whale stranding events that occurred during the 1960s and 1970s in Japan,
56 southern California,”

We removed the reference to southern California and included a reference for the stranding events in Japan

Line 68 change “first beaked whale stranding on Guam” to “first beaked whale stranding recorded on Guam”

Done

Lines 113-117 change “When MFAS packets were
114 detected, they generally occurred in consecutive bouts with fewer than two minutes between
115 sonar packets, followed by “breaks” in sonar activity, which ranged from 15 minutes to nearly 3
116 hours (Table 2). MFAS events were detected on a total of 35 days between 2011 and 2014, with
117 MFAS events lasting from 1 to 18 days (Table 2).”

To

“MFAS events were detected on a total of 35 days between 2011 and 2014, with
117 MFAS events lasting from 1 to 18 days (Table 2). When MFAS packets were
114 detected, they generally occurred in consecutive bouts with fewer than two minutes between
115 sonar packets, followed by “breaks” in sonar activity, which ranged from 15 minutes to nearly 3
116 hours (Table 2).”

Done

Lines 136-146 The statistical test assumed an equal probability of beaked whale availability for stranding thought the time period, but the acoustic detections show otherwise. Unless authors can more convincingly justify assumption of equal probability, better to draw the null hypothesis dates from a

distribution that more closely matches the detections of each species.

The recording effort across all months in all years has not been even, and the apparent seasonal trend in the data at the western recording location may be an artifact of the uneven recording effort. See figures below, which show the average daily presence of Cuvier's beaked whales in bars (left axis), with the recording effort in red asterisks (right axis) for each recording location.

Further, the acoustic detection rates at each site reflect fine scale habitat preferences that likely do not reflect the conditions across the general area. For example, the detection rates for Cuvier's beaked whale were much lower at the east site in all years than the west site, and we have no way of knowing which distribution would be more appropriate for animals that stranded on Guam and Saipan.

As far as we are aware, there is no evidence of large scale spatial movements (e.g. migration between feeding and reproductive areas) in global beaked whale populations. However, there is evidence for high site-fidelity in island associated beaked whale populations around Hawaii (Baird 2019), Southern California (Schorr et al. 2018), and the Bahamas (Claridge and Durban 2012), and animals are known to undergo small-scale movements which could influence the acoustic detection rates at recording locations (MacLeod and Zuur 2005; Schorr et al. 2009, 2014).

Therefore, we believe that it is most appropriate to use a uniform distribution for the null hypothesis dates for potential stranding events throughout the Mariana Archipelago.

Lines 157-159 another indicator of low prob of detecting beaked whales when they are in the area is the low number of minutes/day during days when they ARE detected.

True. Added a statement here to support this: "Another indicator of the low probability of detecting beaked whales in the area is the consistently low number of minutes per day with detections."

Sentence in 200-203 does not make sense. Clarify.

This section has been removed for brevity.

Sentence in 215-218 does not follow. Risk of sonar may be just as high elsewhere.

Absolutely. There may also be similar risks in other places that haven't been measured. We removed the word "particularly" from (Line 287 in revised doc) to remove the comparison to other areas.

In the following section, we also added a line that acknowledges the possibility that similar risks may exist in other regions with similar conditions (Line 326 in revised version).

Line 231 – beaked whale move off range during and after exercise but it is not known if they abandon foraging areas or just shift foraging areas

True. Changed text to: "...or learn to abandon preferred habitat during MFAS operations"

Figure 2 needs to label MFA in bottom cell

Done

References included in the Response to Referees

Au, Whitlow W. L., and Marc O Lammers. 2016. "Listening in the Ocean." In *Modern Acoustics and Signal Processing*, edited by William H Hartmann. Springer New York. <https://doi.org/10.1007/978-1-4939-3176-7>.

Baird, Robin W. 2019. "Behavior and Ecology of Not-So-Social Odontocetes: Cuvier's and Blainville's Beaked Whales," 305–29. https://doi.org/10.1007/978-3-030-16663-2_14.

Baumann-Pickering, Simone, Marie A Roch, Robert L. Brownell Jr., Anne E. Simonis, Mark A McDonald, Alba Solsona-Berga, Erin M. Oleson, Sean M Wiggins, and John A Hildebrand. 2014. "Spatio-Temporal Patterns of Beaked Whale Echolocation Signals in the North Pacific." *PLoS One* 9 (1): e86072. <https://doi.org/10.1371/journal.pone.0086072>.

- Bernaldo de Quirós, Y., A. Fernandez, Robin W. Baird, R. L. Brownell, N. Aguilar de Soto, D. Allen, M. Arbelo, et al. 2019. "Advances in Research on the Impacts of Anti-Submarine Sonar on Beaked Whales." *Proceedings of the Royal Society B: Biological Sciences* 286 (1895): 20182533. <https://doi.org/10.1098/rspb.2018.2533>.
- Claridge, DE, and JW Durban. 2012. "Distribution, Abundance and Population Structuring of Beaked Whales in the Great Bahama Canyon." *Final Technical Report to the Office of Naval Research*. http://www.bahamaswhales.org/news/2009/ONR09_BBES_Dec09.pdf.
- Commander United States Navy Pacific Fleet. 2010. "Southern California Range Complex Environmental Impact Statement," 1–1952.
- D'Amico, Angela, Robert C. Gisiner, Darlene R. Ketten, Jennifer A. Hammock, Chip Johnson, Peter L. Tyack, and James Mead. 2009. "Beaked Whale Strandings and Naval Exercises." *Aquatic Mammals* 35 (4): 452–72. <https://doi.org/10.1578/AM.35.4.2009.452>.
- Díaz-Delgado, J., A. Fernández, A. Xuriach, E. Sierra, Y. Bernaldo de Quirós, B. Mompeo, L. Pérez, et al. 2016. "Verminous Arteritis Due to *Crassicauda* Sp. in Cuvier's Beaked Whales (*Ziphius Cavirostris*)." *Veterinary Pathology* 53 (6): 1233–40. <https://doi.org/10.1177/0300985816642228>.
- Fulling, Gregory L., Philip H. Thorson, and Julie Rivers. 2011. "Distribution and Abundance Estimates for Cetaceans in the Waters off Guam and the Commonwealth of the Northern Mariana Islands 1." *Pacific Science* 65 (3): 321–43. <https://doi.org/10.2984/65.3.321>.
- Klinck, H., S.L. Nieukirk, S. Fregosi, K. Klinck, D.K. Mellinger, S. Lastuka, G.N. Shilling, and J.C. Luby. 2015. "Cetacean Studies on the Mariana Islands Range Complex in September–November 2014: Passive Acoustic Monitoring of Marine Mammals Using Gliders. Preliminary Report. Prepared for Commander, U.S. Pacific Fleet."
- . 2016. "Cetacean Studies on the Mariana Islands Range Complex in March–April 2015: Passive Acoustic Monitoring of Marine Mammals Using Gliders." *Final Report. Prepared for Commander, U.S. Pacific Fleet, Environmental Readiness Division, Pearl Harbor, HI*.
- Kommenou, A., D. Psalla, and A. Drougas. 2017. "Significant Pathological Findings and Possible Causes of Death of Stranded Cuvier's Beaked Whales Due to Anthropogenic Activities in Greece." In *Advances in Technology and Research on Beaked Whales and Antisubmarine Sonar*. Costa Calma, Spain.
- MacLeod, Colin D., and Alain F. Zuur. 2005. "Habitat Utilization by Blainville's Beaked Whales off Great Abaco, Northern Bahamas, in Relation to Seabed Topography." *Marine Biology* 147 (1): 1–11. <https://doi.org/10.1007/s00227-004-1546-9>.
- McCarthy, Elena, David Moretti, Len Thomas, Nancy DiMarzio, Ronald Morrissey, Susan Jarvis, Jessica Ward, Annamaria Izzi, and Ashley Dilley. 2011. "Changes in Spatial and Temporal Distribution and Vocal Behavior of Blainville's Beaked Whales (*Mesoplodon Densirostris*) during Multi-Ship Exercises with Mid-Frequency Sonar." *Marine Mammal Science* 27 (3): 206–26. <https://doi.org/10.1111/j.1748-7692.2010.00457.x>.
- Moretti, David, Len Thomas, Tiago Marques, John Harwood, Ashley Dilley, Bert Neales, Jessica Shaffer, et al. 2014. "A Risk Function for Behavioral Disruption of Blainville's Beaked Whales (*Mesoplodon Densirostris*) from Mid-Frequency Active Sonar." *PLoS ONE* 9 (1): 1–6. <https://doi.org/10.1371/journal.pone.0085064>.
- Munger, L.M., M.O. Lammers, J.M. Oswald, T.M. Yack, and W.W.L. Au. 2015. "Passive Acoustic

Monitoring of Cetaceans within the Mariana Islands Range Complex (MIRC) Using Ecological Acoustic Recorders (EARs). Final Report. Prepared for Commander, U.S. Pacific Fleet, Environmental Readiness Division, Pearl Harbor, HI. Submitted To.”

Naval Facilities Engineering Command Pacific. 2019. “Mariana Islands Training and Testing Activities Draft Supplemental Environmental Impact Statement / Overseas Environmental Impact Statement January 2019.” <https://mitt-eis.com/Documents/2019-Mariana-Islands-Training-and-Testing-Supplement-EIS-OEIS-Draft-Supplemental-EIS-OEIS>.

Oleson, Erin M., Simone Baumann-Pickering, Ana Širović, Karlina P B Merkens, Lisa M Munger, Jennifer S Trickey, and Pollyanna Fisher-Pool. 2015. “Analysis of Long-Term Acoustic Datasets for Baleen Whales and Beaked Whales within the Mariana Islands Range Complex (MIRC) for 2010 to 2013.” *Prepared for U.S. Pacific Fleet Environmental Readiness Office. PIFSC Data Report DR-15-002*.

Schorr, Gregory S., Robin W. Baird, M. Bradley Hanson, Daniel L. Webster, Daniel J. McSweeney, and Russel D. Andrews. 2009. “Movements of Satellite-Tagged Blainville’s Beaked Whales off the Island of Hawai’i.” *Endangered Species Research* 10 (1): 203–13. <https://doi.org/10.3354/esr00229>.

Schorr, Gregory S., Erin A. Falcone, David J. Moretti, and Russel D. Andrews. 2014. “First Long-Term Behavioral Records from Cuvier’s Beaked Whales (*Ziphius Cavirostris*) Reveal Record-Breaking Dives.” *PLoS ONE* 9 (3). <https://doi.org/10.1371/journal.pone.0092633>.

Schorr, Gregory S, Erin A Falcone, Brenda K Rone, and E L Keene. 2018. “Distribution and Demographics of Cuvier’s Beaked Whales in the Southern California Bight.” *Final Report to the US Navy Pacific Fleet Integrated Comprehensive Monitoring Program, Award No. N66604-14-C-0145*.

Southall, B.L., K.J. Benoit-Bird, Mark A. Moline, and David Moretti. 2019. “Quantifying Deep-Sea Predator–prey Dynamics: Implications of Biological Heterogeneity for Beaked Whale Conservation.” *Journal of Applied Ecology*, no. November 2018: 1–10. <https://doi.org/10.1111/1365-2664.13334>.

Tajima, Yuko, Kaon Maeda, and Tadasu K. Yamada. 2015. “Pathological Findings and Probable Causes of the Death of Stejneger’s Beaked Whales (*Mesoplodon Stejnegeri*) Stranded in Japan from 1999 and 2011.” *Journal of Veterinary Medical Science* 77 (1): 45–51. <https://doi.org/10.1292/jvms.13-0454>.

Tyack, Peter L, Walter M. X. Zimmer, David Moretti, B.L. Southall, Diane E Claridge, John W Durban, Christopher W Clark, et al. 2011. “Beaked Whales Respond to Simulated and Actual Navy Sonar.” *PLoS One* 6 (3). <https://doi.org/10.1371/journal.pone.0017009>.

Wensveen, Paul J, Saana Isojunno, Rune R Hansen, Alexander M Von Benda-Beckmann, Lars Kleivane, Sander Van Ijsselmuide, Frans-Peter A Lam, et al. 2019. “Northern Bottlenose Whales in a Pristine Environment Respond Strongly to Close and Distant Navy Sonar Signals.” <https://doi.org/10.1098/rspb.2018.2592>.

Wenz, G.M. 1962. “Acoustic Ambient Noise in the Ocean: Spectra and Sources.” *The Journal of the Acoustical Society of America* 34 (12): 1936–56.